# A Machine Vision-Based Method for Detecting Surface Hollow Defect of Hot-State Shaft in Cross Wedge Rolling

Huajie Fu [1,2], Ying Wang [1,2,*], Xuedao Shu [1,2], Xiaojie Chen [1,2] and Kai Lin [1,2]

1   Faculty of Mechanical Engineering and Mechanics, Ningbo University, Ningbo 315211, China
2   Zhejiang Provincial Key Laboratory of Part Rolling Technology, Ningbo 315211, China
*   Correspondence: wangying5@nbu.edu.cn

**Abstract:** In order to solve the problems of low detection efficiency and safety of artificial surface defects in hot-state cross wedge rolling shaft production line, a machine vision-based method for detecting surface hollow defect of hot-state shafts is proposed. Firstly, by analyzing the high reflective properties of the metal shaft surface, the best lighting method was obtained. And by analyzing the image contrast between image foreground and image background, the most suitable optical filter type in image acquisition was determined. Then, Fourier Gaussian low-pass filtering method is used to remove the interference noise of rolled shafts surface in frequency domain, such as high-light, oxide skin and surface texture. Finally, by analyzing the characteristics of the surface hollow defect area, a defect identification method combining the Otsu threshold method and the adaptive threshold method is proposed to realize the effective extraction of surface hollow defect of rolled shafts. The test results show that the average recognition rate of the method based on machine vision is 95.7%. The results of this paper provide technical support to meet the production requirements of high quality and high performance of cross wedge rolling.

**Keywords:** machine vision; cross wedge rolling; defect detection; surface hollow defect; hot-state shaft

## 1. Introduction

The cross wedge rolling is an advanced technology for the forming of shaft parts, which is widely used in the forming of shaft parts of automobiles, tractors, construction machinery, drilling machinery, coal mining machinery and hardware tools [1–3]. When rolling, the metal billet is heated to about 1150 °C and sent into the mill rolls which distributes symmetrically. Then, the mill rolls rotate and roll the billet into shaft parts with deformed sections. In the above process, the uneven deformation of metal easily leads to some defects on the surface of the formed shaft parts, such as backfin, spiral convex mark and surface hollow [4]. Where the backfin and spiral convex mark can be corrected by finish-cutting. However, the surface hollow defects, as shown in Figure 1, are mostly uncorrectable defects, which are mainly caused by mill roll damage. In order to ensure the surface quality of rolled shaft and the normal operation of the mill roll in the life span, it is necessary to detect the surface hollow defect on the rolled shaft in real time during rolling. At present, human vision are still widely used in cross wedge rolling industry, which has the disadvantages of low efficiency, high omission ratio, high risk factor in heavy machinery and high temperature environment. It is unable to meet the production needs of high efficiency and high performance of cross wedge rolling [5–7].

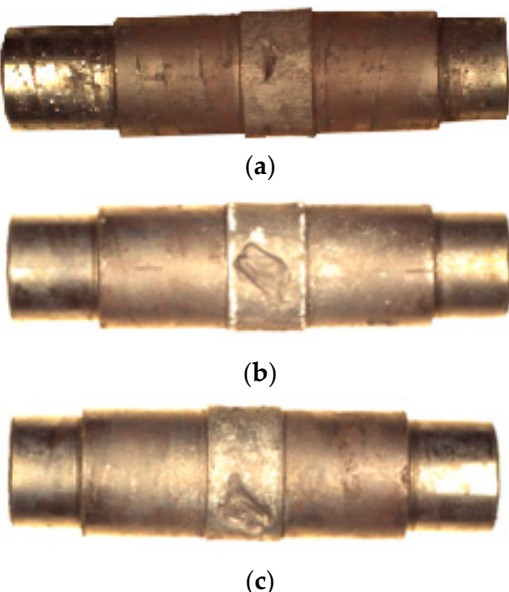

**Figure 1.** The rolled shafts with hollow defect. (**a**) Defect 1; (**b**) Defect 2; (**c**) Defect 3.

With the rapid development of machine vision technology, more and more researchers and engineers use machine vision methods to detect the damage on the mechanical part surface. In terms of the defect detection of curved part surface, S Satorres Martinez [8] presents a machine vision system capable of revealing, detecting and characterizing defects on curved surfaces, In addition, for the defect segmentation, a new adaptive threshold selection algorithm is proposed. Manhuai Lu [9] proposed a bearing surface defect detection and classification method using machine vision technology, it includes a local multi-neural network image segmentation algorithm and a feature selection algorithm, effectively improve the inspection speed and the defect recognition rate of the bearing surface. Qinbang Zhou [10] extracts candidate defect areas from the automobile surface image by a multi-scale Hessian matrix fusion method, then, candidate defect areas are classified into pseudo-defects, dents and scratches by feature extraction (shape, size, statistics and divergence features) and a support vector machine algorithm. Hao Shen [11] develop a machine vision system for bearing defect detection, which proposed system enhances the defects appearances and get controlled image acquisition environment. A series of image processing methods are utilized to inspect the defects, find a common rule on the distribution of projection, and design a simple but effective detection algorithm based on the rule. G Rosati [12] employed a non-flat mirror to illuminate and inspect high reflective curved surfaces. The rays emitted from a light source are conveyed on the surface under investigation by means of a curved mirror. After the reflection on the surface, the light rays are collected by a CCD camera. And the defects will be identified as shadows inside a high brightness image. Iker Pastor-Lopez [13] proposed a new method that detects imperfections on the casting curve using a segmentation method that marks the areas of the casting that may be affected by some of these defects and, then, applies machine-learning techniques to classify the areas in correct or in the different types of faults. The results show that this method obtains high precision rates. Yih-Chih Chiou [14] developed a machine-vision-based system for detecting flaws occurred in cylindrical inner and outer surface defects of polyurethane packaging. Which applied variance of radius inspection method, projection method, coordinate transformation techniques, and normalized grayscale absolute difference inspection method to locate frequently detected defects.

In the above research, the detecting objects are all cold-state. However, the cross wedge rolling shafts are still at high temperature after rolling, which will generate oxide skins and infrared radiation. In addition, due to the presence of dust in the surrounding environment, the imaging noise generated by dust and the complex surface noise of the rolled shafts,

make it different from the traditional surface detection of curved parts. Therefore, this paper proposes a machine vision-based method for detecting surface hollow defect of hot-state shafts in cross wedge rolling. Firstly, lighting and filtering methods are chosen to improve quality of the acquired images. Then, the interference noises are filtered out by analyzing the surface noise of rolled parts in frequency domain. Finally, the defect extraction method is determined through the analysis of characteristics of the surface hollow defect area, and the accuracy of the system is verified through the experiment. This paper lays a foundation for automatic defect detection of shafts in cross wedge rolling.

## 2. Construction of the Machine Vision-Based System

(1)　The schematic diagram of the surface hollow defect detection system for hot-state shaft in cross wedge rolling is shown in Figure 2, and it mainly consists of the following three modules:Image acquisition module. The image acquisition module includes industrial camera and camera lens. The industrial camera is manufactured by Daheng Image Company in Shanghai, China, with the product model is ME2P-2612-4GC-P CMOS. The resolution of the camera is $5120 \times 5120$ pixels and its rectangular chip size ($L \times H$) is 12.8 mm $\times$ 12.8 mm. The product model of the camera lens is DaHeng Image's HN-P-1624-25M-C1.2/1 with focal length $f$ of 16 mm. According to Equation (1), the field of view $\varnothing$ of the camera module can be calculated as 43.6°. Therefore, the shafts will not overflow the image boundary during imaging.

$$\varnothing = 2\arctan\frac{L}{2f} \tag{1}$$

(2)　Lighting module. The lighting module includes two identical LED light bars and background plate. LED light bars are used to provide lighting for the shafts in cross wedge rolling in the image acquisition area, and their lengths are 30 cm. The background plate is made of white aluminosilicate refractory fiber board, which can isolate the high temperature of 1260 °C, and it is mainly used to reduce the noise of images.

(3)　Fixture module. The fixture is a slotted wheel mechanism with telescopic end. The shaft is clamped by controlling the movement of the telescopic end, and the shaft intermittent rotation is driven by controlling the rotation of the groove wheel mechanism. Thus, the overall surface of the shaft image acquisition is achieved.

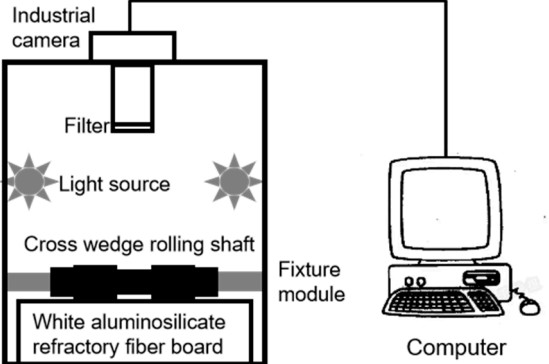

**Figure 2.** Schematic diagram of the surface hollow defect detection system.

## 2.1. Lighting Method

Metal shafts have highly reflective characteristics. When machine vision technology is applied to detect its surface hollow defect, the high-light noise mixed in the collected image is easy to drown useful information such as defects [15–17], which hinders the identification of the surface hollow defect. In order to improve the quality of the acquired image, the

imaging effects of 850 °C cross wedge rolling shaft in two lighting modes, circumferential and axial light source, are compared and analyzed, as shown in Figure 3.

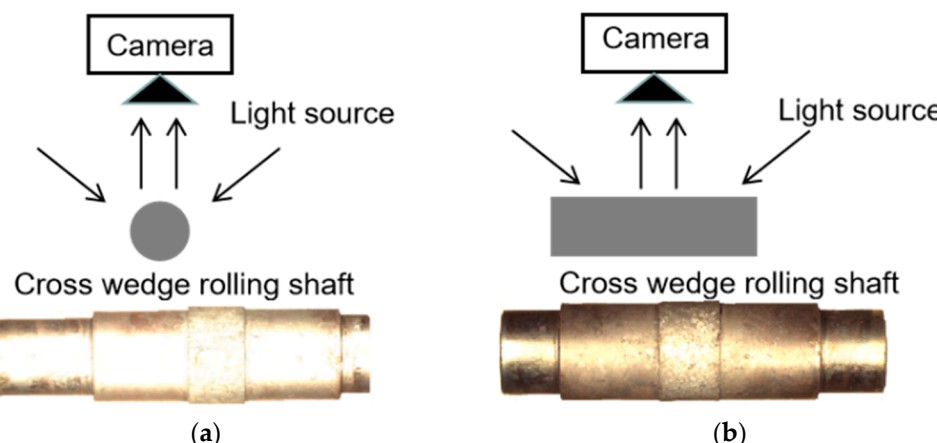

**Figure 3.** Comparison of images acquired by two lighting methods. (**a**) Circumferential illumination; (**b**) axial illumination.

From Figure 3a, due to the influence of the high reflective characteristics of shaft surface, it's easy to form uneven lighting areas and high-light areas when using the circumferential lighting source, which obscuring the shaft surface details. From Figure 3b, the phenomenon of uneven lighting and high-light is significantly reduced when using the axial lighting source, and the shaft surface details can be displayed completely. Therefore, the axial lighting source is the most appropriate, and the LED light bar is placed above the shaft at a 30° angle on both sides.

### 2.2. Light Filtering Method

### 2.2.1. Image Contrast

In the acquired image, the areas covered by the rolled shaft are regarded as the image foreground, and other irrelevant areas are regarded as image background. Before image analysis of the rolled shaft surface, the image foreground should be separated from the whole captured image. And the greater the contrast between foreground and background, the higher the accuracy of the separation result. The Contrast $\Delta V$ is defined as the absolute difference value between the average gray level $V_o$ of image foreground and the average gray level $V_B$ of image background [18].

$$\Delta V = |V_o - V_B| = |\sum_{l=0}^{255} h\frac{n_l}{n^*} - \sum_{L=0}^{255} H\frac{n_L}{n}| \tag{2}$$

where, $h$ is the gray level of image foreground, $n_l$ is the number of pixels with gray level $h$ in image foreground, $n^*$ is the total number of pixels in image foreground; $H$ is the gray level of image background, $n_L$ is the *number* of pixels with gray level $H$ in image background, $n$ is the total number of pixels in image background.

### 2.2.2. Selection of the Optical Filters

Hot-state shafts in cross wedge rolling appear in the form of red heat in dark environment. A lot of heat will emits, making a layer of halo on the surface. Because of the influence of halo phenomenon, the quality of the captured image is low, causing image contrast between the image foreground and the image background is not obvious. The optical filter is always used to absorb some wavelengths in the certain wavebands, to mitigate the influence of halo. Different filters have different transmittance to light of different wavebands. In this paper, blue optical filter BP 465 nm, green optical filter BP 530 nm, red optical filter BP 635 nm and infrared optical filter IR-CUT are selected to test. Image

acquisition was carried out at the same position of 45# steel shaft in cross wedge rolling at 800 °C under the condition of no light. The image acquisition results with different optical filters are shown in Figure 4.

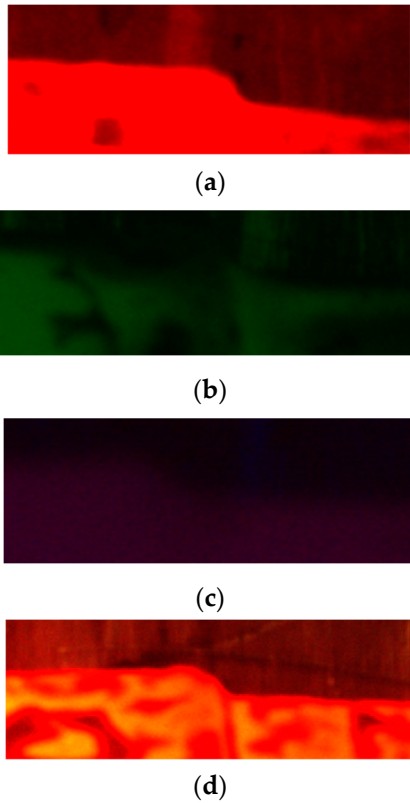

**Figure 4.** Image acquisition results with different optical filters at 800 °C without light. (**a**) Red optical filter; (**b**) Green optical filter; (**c**) Blue optical filter; (**d**) Infrared optical filter.

To determine the optimal filtering method, the maximum image contrast is taken as the index to analyze the influence of different optical filters on the image quality. The temperature range of the rolled shafts are concentrated in 800~900 °C. In this study, the rolled shafts are heat to 800 °C in the furnace and the industrial camera collect the images every 20 °C until the temperature reach 960 °C. Then, segmenting their image foreground and image background separately. By substituting the above two parameters into Equation (2), the image contrast can be calculated. And Figure 5 shows the image contrast results with different optical filter and different temperature. It can be seen from Figure 5 that the image contrast will drops sharply with the increase of temperature when there is no optical filter, and it will changes slowly when the camera equipped with optical filter. As a result, the application of the optical filter can effectively reduce the influence of temperature change for the captured image quality. In addition, the larger the image contrast, the larger the gray value difference between foreground and background, the clearer the contour details and surface texture in the image, which is conducive to accurate image segmentation. Through comparative analysis, the image contrast is largest when the infrared optical filter is applied. As a result, the infrared optical filter IR-CUT is used in the subsequent research.

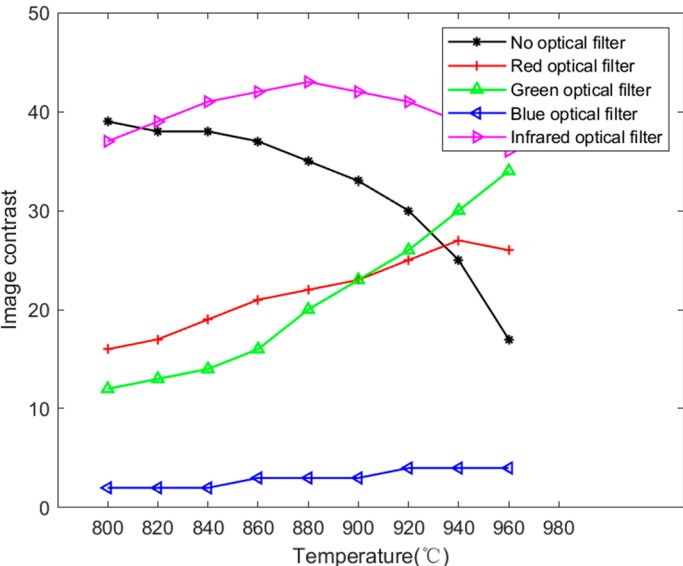

**Figure 5.** Image contrast results with different optical filters and different temperatures.

## 3. Process and Analysis of Surface Hollow Defect Detection

### 3.1. Elimination and Analysis of Rolled Shaft Surface Noise

#### 3.1.1. Fourier Transform

In the process of extracting surface defects, the surface noise in the image will cause great influence. As shown in Figure 1, there are many surface noises such as oxide skin, uneven lighting, step contour and surface texture on the surface of hot-state shaft in cross wedge rolling under strong lighting. Where the oxide skin and uneven lighting easily leads to the staging change of surface gray value. The traditional spatial filtering method is not suitable for the conditions with surface gray value difference, which makes it difficult to extract and identify the surface hollow defect [19,20]. In this paper, transforming the complex images into frequency-domain space by Fourier transform, thus the processing of various noises are more targeted.

The Fourier transform principle states that any continuously measured signals can be represented as an infinite superposition of sine wave signals with different frequencies. As a result, the original signal can be regarded as the superposition of multiple sine wave signals.

Treating the two-dimensional discrete image which size is $M * N$ as a function $f(x, y)$, where $x$ and $y$ are two independent unrelated variable [21,22]. Its Fourier forward and inverse transformation expression can be expressed as:

$$F(u, v) = \frac{1}{MN} \sum_{x=0}^{M-1} \sum_{y=0}^{N-1} f(x, y) e^{-j2\pi\left(\frac{xu}{M} + \frac{yv}{N}\right)} \tag{3}$$

$$f(x, y) = \sum_{u=0}^{M-1} \sum_{v=0}^{N-1} F(u, v) e^{j2\pi\left(\frac{xu}{M} + \frac{yv}{N}\right)} \tag{4}$$

After mapping the image to the frequency domain space by Fourier transform, each area in the frequency domain image will corresponds to a specific frequency domain part in the spatial domain. And the frequency domain distribution of the original image can be obtained [23,24]. Pixels with different gray values in the spatial domain image correspond to different frequency properties, and the target area presents a bright energy distribution in the spectrogram [25]. Therefore, selecting a reasonable filter in the frequency domain space can weaken the corresponding frequency domain energy peak and eliminate the corresponding noise interference.

#### 3.1.2. Low-Pass Filter

Low-pass filter can preserve the low frequency components in the image and remove the medium-high frequency components in the image. Namely, the edge area of the image

can be retained, and the noise interference of oxide skins and surface texture will be removed. The Gaussian low-pass filter has the smooth transition, thus it is selected to filter the original image. And the two-dimensional function expression of the Gaussian low-pass filter is as follows:

$$H(u,v) = e^{\frac{-D^2(u,v)}{2D_0^2}} \tag{5}$$

where, $D_0$ represents its cutoff frequency, $D(u,v)$ represents the distance from the center of the frequency rectangle.

From Equation (4), the inverse Fourier transform function expression of the Gaussian low-pass filter can be obtained:

$$f(x,y) = \sum_{u=0}^{M-1} \sum_{v=0}^{N-1} H(u,v)e^{j2\pi\left(\frac{xu}{M}+\frac{yv}{N}\right)} \tag{6}$$

A Gaussian function is still the Gaussian function through the inverse Fourier transform. Therefore, only its amplitude and variance are changed without affecting the quality of the image after conversion. Gaussian low-pass filter is used to filter the image of shaft in cross wedge rolling, and its image binarization effect is shown in Figure 6.

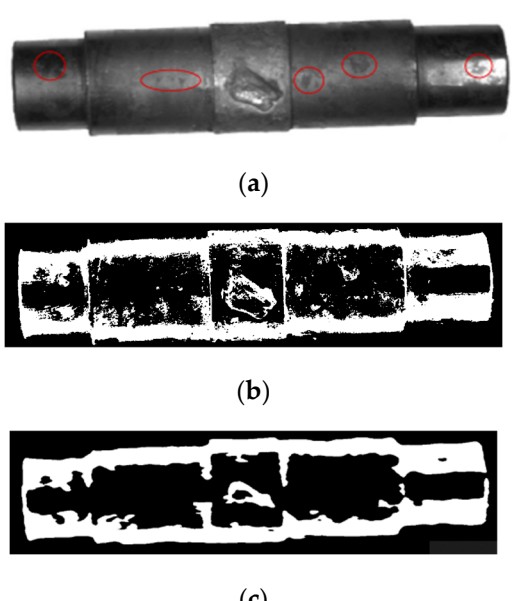

(**a**)

(**b**)

(**c**)

**Figure 6.** Binarization comparison. (**a**) Original image; (**b**) Binarized image; (**c**) Binarized image after filtering.

The part circled in red in Figure 6a is the oxide skins on the shaft surface of hot-state shaft in cross wedge rolling. Compared with Figure 6b,c, it can be seen that the Gaussian low-pass filter can filter most of the oxide skins and surface texture.

### 3.2. Extraction and Analysis of Surface Hollow Defect

After filtering, the image should be segmented in order to extract the surface hollow defect. The Otsu threshold method and adaptive threshold method are two common methods in the image segmentation. The Otsu threshold method uses the global gray-scale average to find the best threshold value, which is sensitive to light. Figure 7a shows the acquired segmentation results by using Otsu threshold method. Where the white area in the shaft contour are two highlight areas of the defect position and the shaft axis position respectively. The adaptive threshold method divides the image into N areas, and the image in different areas will have the corresponding local threshold, which is sensitive to the surface contour. Figure 7b shows the acquired segmentation results by using adaptive threshold method. Where the white area in the shaft contour is step contour, defect contour

and partial uneliminated surface texture. The surface distribution of the hot-state shaft in cross wedge rolling is complex, and the above two threshold processing methods cannot effectively extract the surface hollow defect from many noises.

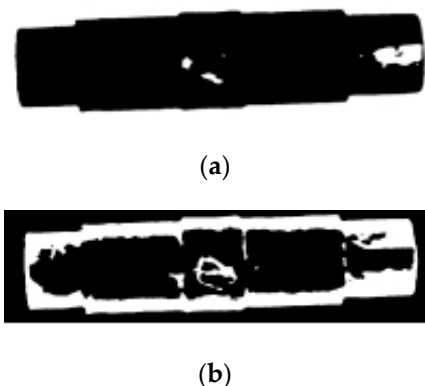

(**a**)

(**b**)

**Figure 7.** Comparison of two threshold methods. (**a**) Otsu threshold method; (**b**) Adaptive threshold method.

By synthesizing the results of above two threshold method, three characteristics of the surface hollow defect area can be obtained: (1) Under the strong lighting, due to the rough surface of the defect area, it is easy to produce diffuse light reflection, which appears as local highlight during imaging. The defect area can be binarized into white by the Otsu threshold method, and the white area is defined as area A; (2) The defect area generally has an obvious edge contour, which can be binary into white by the adaptive threshold method, and the white area is defined as area B; (3) For all types of surface hollow defect, area A is located inside area B. It can be seen from the above that the defect can be judged to be a real defect or interference noise by judging whether the defect area A is inside the area B. Therefore, in this paper, the surface hollow defect is extracted based on the processing results of the Otsu threshold method and the adaptive threshold method. The specific steps are as follows:

After removing the outer contours in Figure 7a,b, polygon fitting and rectangle fitting are performed respectively, and the results shown in Figure 8 can be obtained.

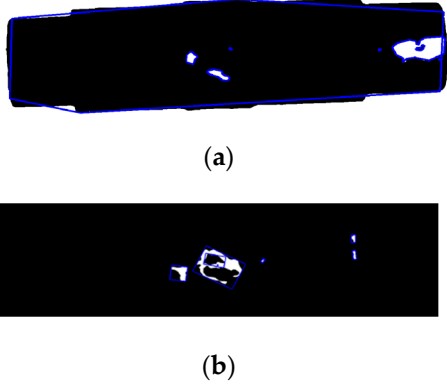

(**a**)

(**b**)

**Figure 8.** Edge fitting. (**a**) Polygon fitting; (**b**) Rectangular fitting.

To judge whether the white area A lies within the white area B, that is, to judge whether all vertices of the polygon fitted by the Otsu threshold method inside the rectangle fitted by the adaptive threshold method. The above judgment process can be conducted by vector cross product, as shown in Figure 9.

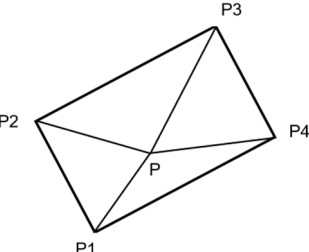

**Figure 9.** Position relationship of inner points of rectangle.

Where $P_1$, $P_2$, $P_3$ and $P_4$ are the four vertices of the rectangle at the bottom, left, top and right, and $P$ is a point on the image. If point $P$ is inside the rectangle, the above points will meet the following equations.

$$f = (P_2 - P_1) \times (P - P_1) * (P_4 - P_3) \times (P - P_3) \geq 0 \tag{7}$$

$$g = (P_3 - P_2) \times (P - P_2) * (P_1 - P_4) \times (P - P_4) \geq 0 \tag{8}$$

Suppose that the number of fitting polygons in Figure 8a is M, the number of vertices of polygon $i$ is $m$, $P_{x,i}$ is the $x$-th vertex of polygon, the number of fitting rectangles in Figure 8b is N, $P_{a,j}$, $P_{b,j}$, $P_{c,j}$ and $P_{d,j}$ are the lowest, leftmost, uppermost and right vertices of the rectangle $j$. If the vertex $x$ of the polygon $i$ is inside the rectangle $j$, from Equations (7) and (8), we can obtain:

$$f(x,i,j) = \left(P_{b,j} - P_{a,j}\right) \times \left(P_{x,i} - P_{a,j}\right) * \left(P_{d,j} - P_{c,j}\right) \times \left(P_{x,i} - P_{c,j}\right) \geq 0 \tag{9}$$

$$g(x,i,j) = \left(P_{c,j} - P_{b,j}\right) \times \left(P_{x,i} - P_{b,j}\right) * \left(P_{a,j} - P_{d,j}\right) \times \left(P_{x,i} - P_{d,j}\right) \geq 0 \tag{10}$$

If all vertices of polygon $i$ are located in rectangle $j$, the $i$-th white polygonal area in the Otsu threshold method and the $j$-th white rectangular area in the adaptive threshold method together constitute the defect area of rolled shafts. It can be expressed as Equation (11):

$$\sum_{x=0}^{m}[f(x,i,j) + g(x,i,j)] = \sum_{x=0}^{m}[|f(x,i,j)| + |g(x,i,j)|] \tag{11}$$

As shown in Figure 10, polygon $i$ and rectangle $j$ are drawn in the original image. The rectangular area is the defect contour area of the rolled shaft, and the polygonal area is the highlight area in the defect.

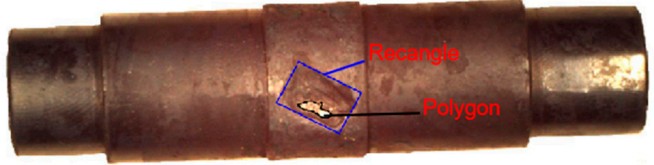

**Figure 10.** Identification of defect area in original image.

Refer to the product requirements of China cross wedge rolling company: if the maximum length of rough surface hollow defect of shaft in cross wedge rolling exceeds 5 mm, finishing cannot eliminate the impact, and it will be classified as unqualified workpiece. In the actual experiment, the billet with 45# material and 30 mm diameter was heated to 1200 °C, and H630III cross wedge mill was used to carry out the cross wedge rolling experiment. A total of 100 shaft billet images were collected, including 30 images without surface hollow defect and 70 images with surface hollow defect. The defect areas were required to be randomly distributed, and its maximum length should be larger than 3 mm. Surface hollow defects were extracted and recognized from the collected shaft images, and the recognition results are shown in Table 1.

**Table 1.** Surface hollow defect identification results.

| Type | Number | Accuracy | | | |
|---|---|---|---|---|---|
| | | Circumferential Illumination + Spatial Filtering Method | Axial Illumination + Spatial Filtering Method | Circumferential Illumination + Low-Pass Filter | Axial Illumination + Low-Pass Filter |
| defective shaft | 70 | 57.1% | 82.9% | 61.4% | 95.7% |
| qualified shaft | **30** | **73.3%** | **86.7%** | **76.7%** | **100%** |

From Table 1, in the test set samples, the overall recognition rate of shaft surface hollow defect is 95.7%. The main reason leading to the error is that in some tests, the surface hollow defects of the captured images are located at the edge position.

## 4. Conclusions

In order to identify the surface hollow defect of hot-state shaft in cross wedge rolling, this paper proposes a machine vision-based detection method, which can effectively realize the rapid and accurate surface defects detection. The specific conclusions are as follows:

(1) Under the strong lighting, the axial lighting source can greatly reduce the high-light noise of the shaft surface; When collecting the shaft image, the infrared optical filter IR-cut can be installed on the capturing camera to make the image foreground and image background have greater contrast, so as to improve the imaging quality of the shaft.

(2) Based on Gaussian low-pass filter to remove the medium and high frequency components in frequency domain, so as to eliminate the interference of surface noises effectively.

(3) The surface hollow defect extraction method based on Otsu threshold method and adaptive threshold method can effectively identify the surface hollow defect of hot-state shaft in cross wedge rolling. After many trials, the average defect recognition rate is up to 95.7%, which proves the effectiveness of this method.

**Author Contributions:** Conceptualization, H.F. and Y.W.; software, H.F. and X.C.; formal analysis, H.F.; data curation, H.F., Y.W., X.S., X.C. and K.L.; writing—original draft preparation, H.F.; writing—review and editing, H.F., Y.W., X.S., X.C. and K.L.; supervision, Y.W.; funding acquisition, Y.W. and X.S. All authors have read and agreed to the published version of the manuscript.

**Funding:** This research was funded by National Natural Science Foundation of China (51975301); Zhejiang Provincial Natural Science Foundation of China (LZ22E050002); Ningbo Science and Technology Major Project (2022Z009).

**Data Availability Statement:** Data is contained within the article.

**Conflicts of Interest:** The authors declare no conflict of interest.

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
