# Peer review of "A Machine Vision-Based Method for Detecting Surface Hollow Defect of Hot-State Shaft in Cross Wedge Rolling"

_metals, doi:10.3390/met12111938_

Round 1
Reviewer 1 Report
The paper presents an application of machine vision method for detecting surface defects in hot-rolled shafts. The novelty and scientific importance of this work is low. There is an abundance of machine vision methods that can be applied in the analyzed cased. These include feature recognition methods and AI-based methods. Authors did not provide insightful analysis and synthesis of the literature and industrial practice. They applied one method without any benchmark. They did not provide definitions of their metrics used in their validation. They did not explain the effects of lighting and image processing parameters in their validation metrics. Some other metrics should be provided in the determination of classification performance. All existing classification methods are neglected by the authors.
The other issue is that the method is used off-line, although current state-of-the-art indicates of using on-line and in-line measurements and verification.
To sum up, the authors presented some industrial value but the presented manuscript is a mere technical report not an insightful scientific paper which adds a value to the scientific community. This also concerns the fact that the author did not discuss their results at all. Mainly because, no discussion is possible with such a limited study.
Reviewer 2 Report
In my opinion, the work is interesting from an engineering point of view, but it has fundamental shortcomings from the scientific point of view.
The presented method should be validated and the results verified. First of all, the obtained results of the image analysis should be verified by other evaluation / measurement methods. This would be the basis for the claim that the method is suitable for assessing surface defects.
The work requires a thorough redrafting and, in its present form, it is not suitable for publication.
Reviewer 3 Report
The article "A Machine Vision-Based Method for Detecting Surface Hollow Defect of Hot-State Shaft in Cross Wedge Rolling" proposes a method for identifying defects in industrial parts at the production stage. This is an urgent problem in general, but the features of the described procedure relate to specific objects and conditions of observation. The article may be of interest as an example of the successful use of a certain set of tools used in machine vision.
As for the text and figures, the description of the main procedures is quite clear, and the figures fully illustrate the entire process of defect detection. The conclusions correspond to the results obtained in the article.
Minor comments.
1. Section 2.2.2., the first paragraph:
I believe that optical filter BP465nm is blue, and optical filter BP635nm is red
2. Eq.2:
right ‘module bar’ is missing;
are h and H equal or proportional to l and L? If the answer is Yes, it is better to show this in the equation.
Round 2
Reviewer 1 Report
Dear Authors,
Thank you very much for improving your paper. There are still some comments that require action. In its current state, the paper
1. "By analyzing the characteristics of cold-state curved surface and hot-state curved surface, it is found that the cold-state curved surface vision detection
method is not suitable for hot-state curved surface vision defect detection. Therefore, the "a machine vision-based method for detecting surface hollow defect of hot-state shaft in cross wedge rolling" was proposed."
The authors did not respond sufficiently to my comment related to the multitude of existing machine vision/feature detection techniques (e.g. SIFT, SURF, ORB, AKAZE, ANN-based etc.). If the authors would like to emphasize that they are analyzing the defects on hot object than they should stress more on problems related to vision inspection of the hot components.
2. Still no comparison is given to other methods. Thus, the limitation of this study is high.
3. No discussion was provided even though it is strongly recommended for original research paper.
To sum up, the paper was not improved significantly to be accepted as a research paper. It is still a technical report.
Round 3
Reviewer 1 Report
Thank you for improving your paper.